# A complex systems model of breast cancer etiology: The Paradigm II Model

**Robert A. Hiatt**[1,2]*, **Lee Worden**[3], **David Rehkopf**[4], **Natalie Engmann**[5], **Melissa Troester**[6], **John S. Witte**[4], **Kaya Balke**[2], **Christian Jackson**[4], **Janice Barlow**[7], **Suzanne E. Fenton**[8], **Sarah Gehlert**[9], **Ross A. Hammond**[10], **George Kaplan**[11], **John Kornak**[1], **Krisida Nishioka**[12], **Thomas McKone**[13], **Martyn T. Smith**[14], **Leonardo Trasande**[15], **Travis C. Porco**[1,3]

1 Department of Epidemiology and Biostatistics, School of Medicine, University of California San Francisco, San Francisco, California, United States of America, 2 Helen Diller Family Comprehensive Cancer Center, University of California San Francisco, San Francisco, California, United States of America, 3 Francis I. Proctor Foundation for Research in Ophthalmology, University of California San Francisco, San Francisco, California, United States of America, 4 Department of Epidemiology and Population Health, Stanford University School of Medicine, Stanford, California, United States of America, 5 Genentech, Inc. South San Francisco, San Francisco, California, United States of America, 6 Department of Epidemiology, University of North Carolina at Chapel Hill, Chapel Hill, North Carolina, United States of America, 7 Zero Breast Cancer (retired), San Rafael, California, United States of America, 8 Division of the National Toxicology Program, National Institute of Environmental Health Sciences, National Institute of Health, Research Triangle Park, North Carolina, United States of America, 9 Suzanne Dworak-Peck School, University of Southern California, Los Angeles, United States of America, 10 Brown School, Washington University, St Louis, Missouri, United States of America, 11 University of Michigan (retired), Ann Arbor, Michigan, United States of America, 12 School of Law, University of California, Berkeley, Berkeley, California, United States of America, 13 School of Public Health, University of California, Berkeley, (Emeritus), Berkeley, California, United States of America, 14 Division of Environmental Health Sciences, School of Public Health, University of California, Berkeley, Berkeley, California, United States of America, 15 Department of Pediatrics, NYU Grossman School of Medicine, New York City, New York, United States of America

* robert.hiatt@ucsf.edu

**Data Availability Statement:** Data for this model was accessed from publicly available data sources: 1. U.S. Census, Decennial Census data, 2010. https://data.census.gov/ 2. The California Cancer Registry, California Department of Public Health.

## Abstract

### Background

Complex systems models of breast cancer have previously focused on prediction of prognosis and clinical events for individual women. There is a need for understanding breast cancer at the population level for public health decision-making, for identifying gaps in epidemiologic knowledge and for the education of the public as to the complexity of this most common of cancers.

### Methods and findings

We developed an agent-based model of breast cancer for the women of the state of California using data from the U.S. Census, the California Health Interview Survey, the California Cancer Registry, the National Health and Nutrition Examination Survey and the literature. The model was implemented in the Julia programming language and R computing environment. The Paradigm II model development followed a transdisciplinary process with expertise from multiple relevant disciplinary experts from genetics to epidemiology and sociology with the goal of exploring both upstream determinants at the population level and

Cancer Inquiry System, 2005-2009. https://explorer.ccrcal.org/ 3. California Health Interview Survey. CHIS 2005. Adult Public Use Data Files. Los Angeles, CA. https://healthpolicy.ucla.edu/chis/data/Pages/GetCHISData.aspx 4. NCHS. National Health and Nutritional Examination Survey, 2007-2012. https://wwwn.cdc.gov/nchs/nhanes/continuousnhanes/default.aspx.

**Funding:** This work was supported by the California Breast Cancer Research Program (20ZB-8303 to R.A. Hiatt). The funder had no role in study design, data collection and analysis, decision to publish, or preparation of the manuscript.

**Competing interests:** No authors have competing interests.

pathophysiologic etiologic factors at the biologic level. The resulting model reproduces in a reasonable manner the overall age-specific incidence curve for the years 2008–2012 and incidence and relative risks due to specific risk factors such as BRCA1, polygenic risk, alcohol consumption, hormone therapy, breastfeeding, oral contraceptive use and scenarios for environmental toxin exposures.

## Conclusions

The Paradigm II model illustrates the role of multiple etiologic factors in breast cancer from domains of biology, behavior and the environment. The value of the model is in providing a virtual laboratory to evaluate a wide range of potential interventions into the social, environmental and behavioral determinants of breast cancer at the population level.

## I. Introduction

Complexity theory and complex systems thinking supports scientific exploration of the causes of disease that go beyond simple additive models and reductionist approaches to knowledge acquisition [1–5]. Reductionist approaches have generated understanding on specific relationships based on an exposure-outcome model, but it is becoming increasingly clear that the real world is more complex and other approaches are needed by epidemiologists and other population scientists to capture properties of emergence, interacting feedback loops, and adaptation to change over time [6,7]. Agent-based models are one type of dynamic systems model approach to such complexity because they consist of heterogeneous individual entities (agents) that can interact and change over time in response to other agents and environmental exposures [8] and produce observations that one might not expect from a detailed (reductionist) examination of the individual agents themselves [4,6,9,10]. They can give life to the oft used quote from Aristotle that 'the whole is more than the sum of its parts'. In this paper we describe our use of an agent-based model to help understand the complexity of breast cancer etiology.

Breast cancer is the most common female cancer worldwide and the second leading cause of death from cancer (after lung cancer) in the United States [11,12]. In 2022 there was an estimated 287,850 new invasive female breast cancer cases and 43,250 deaths [11,12]. Consequently, it is not surprising that there has been an enormous amount of research done on this cancer to understand its origins and how to prevent and treat it successfully. This corpus of research has created a very complex picture of breast cancer etiology, to say nothing of the health care procedures developed to diagnose and treat it. The research has long since gone past any effort to find a single independent cause and instead has built an understanding of its critical dependence on endocrine and reproductive factors over the life course as well as the effects of external exposures to a large variety of agents and circumstances [13,14]. These factors work at multiple levels from biologic to environmental and societal operating over time [15] to result in markedly different incidence rates by race, ethnicity, socioeconomic status, region and country. This complexity has made breast cancer ideally suited to complex systems modeling studies, given the non-linear interaction between all the relevant causative factors in women over their life course [16].

Models of breast cancer risk have taken several approaches. Some of the best known models, such as the Gail model [14,17], attempt to predict risk at the individual level given characteristics of the individual, whereas others have used more epidemiologic data to enhance predictive abilities using detailed cohort data such as the Rosner-Colditz model and its

enhancements [18–21] and the model of Tyrer, Duffy and Cuzick [22]. The Cancer Intervention and Surveillance Modeling Network (CISNET) has produced six models since 2000 when funding from the National Cancer Institute began. Their collaborative output has been substantial over the years and their influence on policies for screening and treatment have had a notable impact on guidelines. Their purpose has always been to evaluate the impact of cancer control interventions on population trends in breast cancer incidence and mortality and to project future trends [23,24]. Many other models of breast cancer have focused on one aspect or another of the development of the tumor, the influence of genetic predisposition (e.g., BRCA1&2 or the path to metastasis [25]. These other models are important for understanding breast cancer pathobiology and for clinical and personal decision-making, but do not inform interventions and policy in population health.

The rationale for our interest in developing the Paradigm II (PII) model was to be useful in supporting prevention measures at the population level prior to diagnosis and treatment. We undertook the development of such a model that accounts for etiologic factors, not only in breast cancer risk behaviors and biology, but also more "upstream" factors in the social, built (i.e., man-made) and toxicological environment. Mechanistic models are important because they provide an avenue to understanding how processes at the individual level generate observed epidemiological patterns at the population level. Our intent is that the Paradigm II model can be used to test population health interventions *in silico* because this approach lends itself to understanding relative and synergistic contributions of inputs at multiple levels.

The increasing availability of large, heterogeneous data sources together with increasing computer power makes it possible to construct comprehensive population models for chronic disease in general and breast cancer in particular.

We have published two iterations of a conceptual framework on which this agent-based model is based [26,27]. We chose the name, Paradigm Model, to reflect our sense that this approach to understanding causal factors for breast cancer reflects an important shift in thinking from more traditional linear approaches that include one or a small number of etiologic factors [4]. In the current revision of the Paradigm model, begun in 2015, 96 variables are included in four domains of social, environmental, behavioral and biologic determinants [26,27]. The Paradigm II (PII) Model was designed mechanistically and mathematically with a reduced number of etiologic variables, which while large make the model more manageable. This model begins with a classical branching process model at the cellular level [28], from which individual-level disease states are derived. Population-level states are then formed as a distribution over the individual level states [29].

Our goal in developing this model was to explore the role of more upstream determinants at the population level while illustrating the contribution to breast cancer causation of multiple factors at different levels of biologic organization [30]. Conceptually we approached this inquiry from the perspective of population health and 'convergence science' by taking a transdisciplinary approach to framing a common question (i.e., the etiology of breast cancer) that required input from multiple disciplines [31]. Here we report on the development of the PII model and a description of its characteristics. We intend that future uses of the model will allow us to explore specific questions including: (1) health inequities by income and education (including, for example, the effect of policies such as the earned income tax credit), (2) the effect of obesity-related interventions at different stages of life, and (3) the effects of environmental chemical exposures.

## II. Methods

We have previously described the process for developing a conceptual model of the complex etiology of breast cancer [27]. As described we took a conscious transdisciplinary approach

[32–36] to the problem and assembled a multidisciplinary 15 person panel with individual expertise in epidemiology, genetics, breast cancer biology, toxicology, biostatistics, population health, mathematical and agent-based modeling, and advocacy. The panel convened three times a year for two years to discuss issues, compile the most up to date literature and debate various challenging aspects of both the conceptual and mathematical model. Individual meetings with team members and consultations occurred between the full panel meetings to address specific questions.

Variables for the model were selected based on current knowledge as assessed by the expert panel members and a literature review that sought to identify all recent and relevant systematic reviews and large high quality studies where systematic reviews were not available. Inclusion and exclusion decisions as well as criteria for both the strength of association and the quality of the data were established and followed as has been previously described [27].

PII is an agent-based microsimulation model of the *physiology* and *population distribution* of breast cancer for the 18,736,126 women in the state of California in 2010, connecting patterns across scales from the gradual transformation of human tissue to statewide incidence patterns. Simulated individuals were tracked from birth to death, modeling explicitly the growth and loss of breast tissue and progressive transformations of tissue through a series of classes potentially leading to cancer. Conceptual and unobservable, these classes do not correspond to life course events, but to internal developments within cells that are milestones on the road to cancer. Exposures relevant to growth, cell loss, and transformation of tissue were distributed according to population level data, including the U.S. Census, the California Cancer Registry, a component of the National Cancer Institute's Surveillance, Epidemiology, and End Results (SEER) program [37], the California Health Interview Survey (CHIS) [38], and the National Health and Nutrition Examination Survey (NHANES) [39]. We modeled the demographic distribution of California in detail allowing close examination of the relationship between population statistics and breast cancer incidence. Our model was a stochastic individual-based model with hidden (unobserved) pre-cancer and cancer classes of transformed cells evolving over time. (See S1 Appendix).

The model was implemented in the Julia programming language, a relatively new computing environment that is optimized for high computational efficiency and rapid, flexible software development (Version 0.5.1) [40]. Data were pre- and post-processed using the R computing environment (R Foundation for Statistical Computing, Vienna, Austria; Version 3.6.1) [41].

## II.A. Physiology model

The physiological layer of the PII model simulated the transformation from healthy dense breast tissue, which reflects the breast tissue at risk better than total breast tissue [42–44]. This tissue is transformed through a series of intermediate states to the potential emergence of a malignancy and its detection (Fig 1). Focusing on etiology, we were interested in the incidence of breast cancer, not in its clinical presentation, treatment and survival. Transformation was modeled as a stochastic process in which tissue at multiple classes of transformed cells could coexist in a single body. Rates of tissue growth, transformation, and cell loss were affected during each modeled individual's life course by genetic background of both high and low penetrance gene mutations, developmental and reproductive events of menarche, parity, and menopause and then behaviors and exposures to breastfeeding, exogenous hormones, chemical toxicants, obesity (i.e., body mass index (BMI)), alcohol consumption, and smoking. Many more factors from the social, built and chemical environment were in the conceptual model [27] and are characteristics of the base population. However, in this version of the

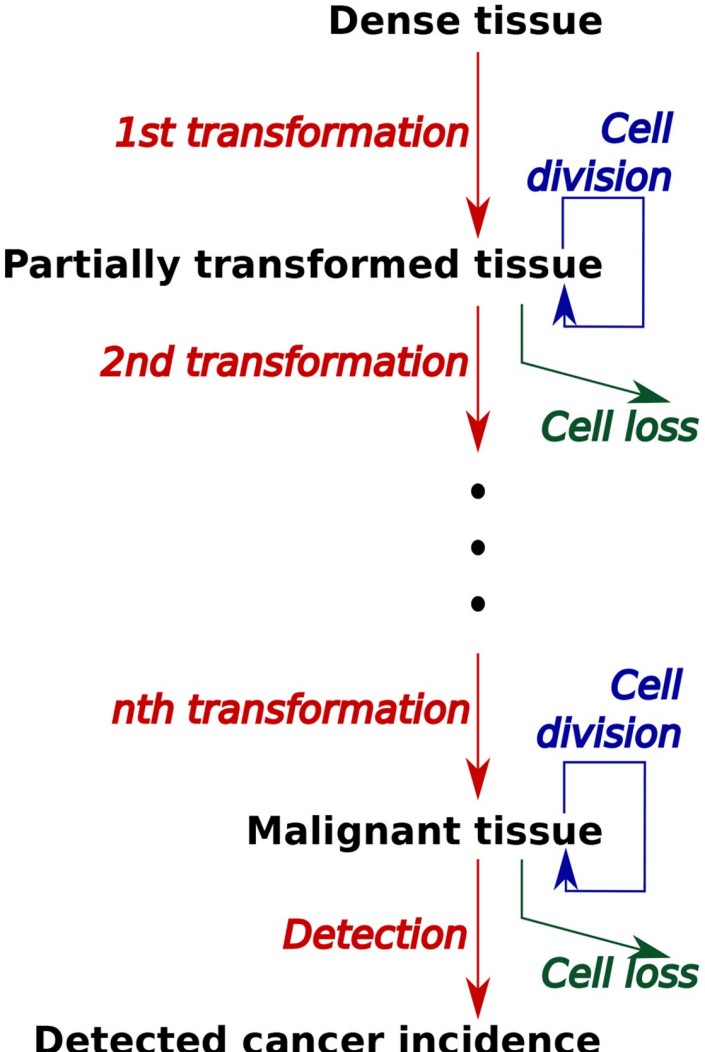

**Fig 1. Progression of tissue through transformations to malignancy in model.** At each stage of transformation, tissue is acted on by processed of cell division (tissue growth), removal, and transformation to the next stage. Different classes of damage occur at different stages of progression. Individuals with malignant tissue are included in incidence statistics by a process of detection.

mathematical model only this subset of factors was included because they were judged by our transdisciplinary team as most critical to breast cancer etiology. Rates of detection of malignant tissue were affected by age-dependent screening rates with mammography. When malignant tissue was detected in a modeled individual, the individual was included in population-wide cancer incidence statistics.

**II.A.1. Individual-based simulation.** The rate parameters of the branching process were modified by hormone levels during the different life stages modeled by Pike [45], and by breastfeeding behavior and other exposures such as alcohol consumption, listed above. This involved the idea of a transformation of the time axis using "tissue age", which advances at a time-varying rate. Hazard was closely related to the fifth to sixth power of tissue age observed in patterns of incidence from registry data [37]. We modeled the progression of tissue to malignancy using a multi-stage model based on the classic multi-stage theory of Armitage and

Doll [46]. Demographic distributions of age at menarche and other reproductive milestones came from NHANES or were specified directly [39].

Agents were introduced at birth and advanced in age each year. Age was not directly involved in the carcinogenesis process, but events such as menarche, child bearing, and menopause occurred as a function of age, and variables such as breast density and tissue growth rate (reflecting the impact of estrogen exposure) depended on the agent's age as pre-menarche, pre-menopausal or post-menopausal.

**II.A.2. Rates of transformation.**   Transformation of healthy dense breast tissue to modified or damaged tissue, and from one pool or class of modified tissue to the next, occurred at a constant rate. We considered dense breast tissue as the target for transformation in the model even though we recognize that there are effects on carcinogenesis from surrounding stromal tissue [47]. The number of transformations, affects the shape of the age-incidence curve matched to reported incidence data [28]. The log-slope of incidence observed in the California data was found to be in between those produced by a model with five transformations and one with six transformations, when transformation rates are homogeneous. We fine-tuned the slope by the following logic. If we were to model six transformations and increase the rate of the final transformation step to near infinity, independent of the other steps, the result would be effectively the same as a five-transformation model. It follows that increasing the rate of the sixth step to a large but not infinite value must produce an intermediate slope. Consequently we used a six-transformation model and calibrated its transformation rates by adjusting the rate for the first five steps to produce cancer incidence at approximately the right ages and adjusting the rate for the final step separately to produce a slope comparable to the observed incidence. The classes of cell transformation were introduced into the model to fit observed age-specific incidence rates. Transformation rates were modified by smoking, alcohol use, and BRCA1 and by low-penetrance genetic variants as discussed in more detail below.

**II.A. 3. Growth rates.**   The rates of growth of each class of tissue varied throughout the life course according to the function developed by Pike et al. [45]. We began with a baseline growth rate assumed to be the same for all classes of modified tissue except the final class. Growth rates were modified according to age dependent developmental events, using the multipliers estimated by Pike et al.[45]: 0 before menarche; 1 from menarche to the first full-term birth; 2.2 the year of the first full-term birth, and subsequently 0.70 until menopause; and 0.105 after menopause.

The growth rate of cancerous tissue was assumed much greater than that of the preceding class and was calibrated to produce timing and overall magnitude of incidence comparable to that observed. Growth rates were modified in ways discussed below by BMI and hormone exposure from hormone therapy and oral contraception.

**II.A. 4. Cell loss rates.**   Rates of cell loss were slower for later classes of modified tissue, calibrated roughly linearly from a baseline rate for once-modified tissue to 0.48 times that for pre-cancerous tissue, and 0.25 times the baseline for cancerous tissue. All cell loss rates were elevated by a factor of 100 during years of breastfeeding, i.e., in model individuals who breastfed, in the year after each birth [48]. Loss rates were also elevated at very advanced ages, to reflect a drop in age-specific incidence observed in California (and national) incidence data. The age of onset and magnitude of this decrease in incidence were calibrated to California age-specific incidence data [37].

**II.A.5. The pool of susceptible tissue.**   The transformation of healthy dense tissue to the first pool of modified tissue occurred at the same per-capita rate as the accumulation of subsequent transformations, in proportion to a quantity of dense breast tissue that was susceptible to the first transformation process. All quantities of breast tissue were modeled in abstract units of volume, in order to capture the interactions between these pools of tissue without

reflecting the true numbers of cells or grams of tissue. Accordingly, the modeled pool of dense tissue was considered to be a proxy for all the breast tissue of an individual calibrated to yield the needed incidence pattern.

Change in dense breast tissue over the life-course was modeled using an idealized tissue-volume dynamic in which the volume rose from zero at menarche to a constant baseline volume (in abstract units of volume), declined by 5% at menopause, which was modeled as a point in time, and declined by 1% at each year after menopause. The baseline post-menarche dense tissue volume was modeled as constant across all individuals, and was a parameter calibrated to the age-specific incidence curve reported by the California Cancer Registry [37]. Mammographic density is associated with a markedly increased risk of invasive breast cancer [49]. To a first order, breast density was considered as a function of dense volume and BMI [50]. Breast density was considered to be modeled by the above effect of dense volume and the effects of BMI [42], since breast density itself was not available to be included in the model.

**II.A.6. Detection.** Cancerous tissue was detected when it reached a threshold magnitude in the absence of screening, and at a smaller threshold in years when screening was done. Thresholds were calibrated by simulating the Canadian National Breast Screening study [51–53], to produce comparable levels of detected cancer at one year from introduction of screening at age 40 years and at two to five years from introduction of screening in populations aged 40–49 and aged 50–59.

## II.B. The population model

The population layer of the PII model used individual-level data from NHANES [39] and racial/ethnic and education level population data reported by the California Department of Finance from the 2010 census [54] to distribute demographic variables across the simulated population accurately. These variables include joint distribution of ages at menopause, menarche and childbirth, breastfeeding, smoking, alcohol use, BMI, education, and race/ethnicity. BRCA1 mutations were distributed using recorded prevalence by race/ethnicity [55,56] and a polygenic risk score summarizing low-penetrance genetic exposures was assigned according to its reported population-wide distribution [55]. Due to the disproportionate representation of white subjects in the studies used, we assumed the distribution of risk in non-white subjects to be comparable though it may be driven by a different distribution of underlying genetic variants. CHIS 2005 data on screening behavior was used to model frequency of screening per individual according to race/ethnicity and rural/urban location [38]. Annual survival by age was drawn from the California Cancer Registry [37].

Unobservable rate constants in the model were calibrated to produce age-specific incidence comparable to that reported by the California Cancer Registry for the State of California (2008–2012) [37], and to reproduce reported risks for specific exposures, as detailed below. Quantitative calibration criteria are discussed in the context of individual variables, below. The life courses of 1,000,000 individuals were simulated to generate model incidence over the synthetic population described above. Results described below are reported from this population, unless noted otherwise. The cancer screening, hormone therapy, and oral contraception statistics are generated in separate population simulations constructed to replicate the outcomes of randomized controlled trials, as described below, and sensitivity analysis was performed by a sample of simulations of synthetic populations and controlled trials with modified parameters, as described below.

**II.B.1. Genetic exposures.** High-penetrance genes were modeled using the distribution and effect of the BRCA1 gene [55] which we considered illustrative of the contribution of germline mutations. BRCA1 confers a slightly higher risk of breast cancer and is associated

with more aggressive disease compared to BRCA2. BRCA 1 prevalence was assigned to random individuals at a rate dependent on race/ethnicity. The effect of BRCA 1 was modeled in multiple ways for comparison: 1) the presence of damaged cells at birth (early or late classes); 2) the increased rates of accumulation of damage; 3) the reduced removal of damaged tissue; and 4) the increased growth rate of damaged tissue. Results were compared in order to select one of the four choices based on its ability to reproduce the effect of BRCA1 on age specific risk. The effect of BRCA1 in the model was calibrated to approximate a breast cancer risk of 57% by age 75.

Low-penetrance genes were modeled using a polygenic risk score [57] that summarized the effect of many common genetic variants on breast cancer risk. A polygenic risk score was assigned to individuals using a normal distribution as described in Mavaddat et al [57]. Due to the disproportionate representation of white subjects in the studies used, we assumed the distribution of risk in non-white subjects to be comparable though it may be driven by a different distribution of underlying genetic variants. The polygenic risk score was assumed to **correspond** to an alteration in the individual's physiology in the same way as the BRCA1 variant, the risk score being used as a multiplier on the rate constants chosen to model BRCA1, with a scaling constant to be determined by calibration to the resulting cancer risk outcomes reported in [57].

**II.B. 2. Obesity and body mass index.**   Obesity was modeled using BMI values that were sampled from the NHANES population [39] and transformed to the four World Health Organization standard categories: Underweight (less than 18.5), Normal (18.5–24.99), Overweight (25–29.99), and Obese (30 or more). We were concerned about the effect of obesity on elevated post-menopausal cancer incidence and not on the protective effect of the much smaller number of pre-menopausal cancers, so we modeled its effect only in the postmenopausal years determined for each woman based on the model. We used data from large systematic reviews to assign increased risk of 12% for each five point increase in BMI [58] and a 20–40% increase for obese women compared to normal weight women [59]. In our model, BMI category membership was transformed into a multiplier for the rates of tissue growth, starting at the time of menopause reflecting the understanding that BMI increases breast cancer risk.

**II.B.3. Alcohol use.**   Alcohol exposure was also sampled from the NHANES population's 'drinks per year' variable [39]. A 'drink' is defined as a 12oz beer, 5 oz. of wine or a 1.5 oz shot of liquor. Model individuals' tissue growth rates were amplified in proportion to the number of drinks per year. Estimates of the effect of alcohol were taken from a meta-analysis of pooled data from 118 studies that reported that i that light drinkers have a slightly increased (1.04-fold higher) risk of breast cancer, moderate drinkers (1.23-fold higher) and heavy drinkers (1.6-fold higher) compared with nondrinkers [60]. Moderate drinking is defined as up to 1 drink a day for women, and heavy drinking as 4 or more drinks a day or 8 or more a week. The Paradigm model was calibrated to reproduce a similar magnitude of risk increase for moderate drinkers.

**II.B.4. Oral contraception.**   NHANES reported ever use and starting and stopping ages for oral contraceptive (OC) use [39]. Because of gaps in the data we did not use the raw values present in the NHANES data set, instead aggregating their joint probability density, and drawing starting and stopping ages jointly for each model OC user. Due to small subpopulation sizes in NHANES, we used an aggregated statistic to describe OC starting and stopping ages. Starting ages were taken from individuals who began OC after menarche, and stopping ages from individuals who stopped before menopause. Starting and stopping ages were each aggregated into four quantiles. For model individuals, starting and stopping ages, random draws were taken from the joint distribution of quantiles, and the median starting and stopping ages in the quantiles drawn were used as the model individuals' starting and stopping ages. OC use was assumed to increase hormone levels and thus tissue growth rates by a fixed factor during

the ages of exposure. The magnitude of this factor was calibrated to reported relative risks associated with OC exposure [61].

**II.B.5. Breast feeding.** Breastfeeding is modeled as protective. The population distribution of breast feeding is modeled using the "ever breastfed" variable of the NHANES data set [39]. Individuals who breastfeed are modeled as breastfeeding during the year following each birth. Breast feeding is modeled as a constant factor increase in tissue removal rate for all classes of transformation, calibrated to produce a relative risk at age 75 of roughly 0.76 [62].

**II.B.6. Hormone therapy.** Hormone therapy (HT) was widely used to treat symptoms of menopause up until the time following the publication of results of the randomized Woman's Health Initiative in 2002 that showed no benefit for heart disease and an increased risk of breast cancer [63]. In our model we used data from NHANES (1988–94) before 2002 that recorded individuals who used combined estrogen/progestin HT, and their starting and stopping ages [64]. Because of gaps in the data we did not use the exact starting and stopping ages reported for each NHANES individual, and instead collected the HT age data into a summarized distribution of stopping ages conditional on starting age. Due to small subpopulation sizes, we aggregated starting age and duration of use into four quantiles each.

Each modeled HT user was assigned a duration of HT use by drawing a quantile of duration of use from the distribution of these quantiles given the quantile of the starting age that was assumed equal to the age of menopause. The median of the quantile drawn was used as the model individual's duration of use. Individuals exposed to HT were assumed to experience higher levels of hormone exposure than would otherwise occur after menopause, modeled as a high tissue growth rate, until their age of stopping HT. Relative risks associated with HT were taken from the Oxford Collaborative Group on Hormonal Factors in Breast Cancer [65].

**II.B.7. Smoking.** Smoking behavior was sampled from the NHANES population's 'ever-smoke' variable [39]. Individuals with a value of 1 for this variable were assumed to be smokers throughout their life-course. Smokers' rates of transformation of tissue were accelerated by a constant factor, whose magnitude was calibrated to produce a population-wide relative risk at age 75 comparable to that observed based on an average starting date of 15 years. Nearly 90% of adult smokers report having started before age 18 years [66].

**II.B. 8. Chemical exposures.** Rather than model the potential effects of a number of chemical toxicants on breast cancer etiology [27], we examined several types or classes of chemical-exposure scenarios based on literature that documented effects from exposures *in utero* [67–73], exposures by endocrine disrupting chemicals that lowered the age of pubertal transition [69–71] and genotoxic exposures during adult life [68,72,73]. The effects of chemical exposures were modeled by exploring the effects of different classes of damage at different classes of progression as illustrated in Fig 1. These were for exposures *in utero* 1) prenatal exposure, introducing some damaged tissue at birth, and 2) prenatal exposure that alters programming by increasing transformation or other rates; for exposures that lower the age of puberty 3) a brief increase in accumulation of the first class of genetic (somatic) damage, in the 20th year only, 4) a persistent increase in accumulation of the first class of damage, from age 20 to 29, 5) each of the above exposures, but affecting the 5th class of damage, and for genotoxic exposures in adult life 6) increase in estrogen-like exposure, i.e. accelerated growth of damaged tissue,. Each of these scenarios was modeled by simulating an idealized population of individuals with identical life histories, divided into a control group and a experimental group for comparison. Each of these individuals experienced menarche at age 12, menopause at age 45, had no children, had no BRCA genes and polygenic risk score of 0.69, did not smoke, and had normal BM.

**II.B.9. Screening.** Screening frequency varies with race/ethnicity, and between rural and urban populations and is important because it affects the probability of detection of a given

breast cancer. We used CHIS [38] to estimate rural/urban proportion as a function of race/ethnicity and assigned model individuals as rural or urban using those frequencies. Frequency of screening by race/ethnicity for rural and urban populations was also taken from CHIS, and we used that distribution of values to assign frequency of screening for each model individual given their rural/urban assignment. The frequency was the number of screenings a survey respondent reported in the last 7 years, from 0 to 7. We used the results of the Canadian National Breast Screening Study [51–53] to calibrate the relative risks generated by the model in women aged 40–49 and 50–59, at 1–5 years from the introduction of annual screening in the treatment group in simulation of the Canadian study. Tumor size thresholds for detection with and without screening and the growth rate of malignant tissue were adjusted to calibrate the result of this experiment in simulation.

### II.C. Sensitivity analysis

Sensitivity analysis was performed using a Latin Hypercube sample of 100 combinations of parameter values perturbed from the values attained by calibration. In each case, 100,000 individual life courses were simulated using the California synthetic population model and in the simulated screening, hormone therapy, and oral contraception trials described above. The sensitivity of outcomes to parameter values was estimated from the model results by linear regression.

## III. Results

### III.A. Physiology model

The age distribution for women in California for the year 2000 is given in Fig 2. This data forms the basis for the simulation of breast cancer and associated risk factors and was created by backfitting to births based on the population distribution in 2000.

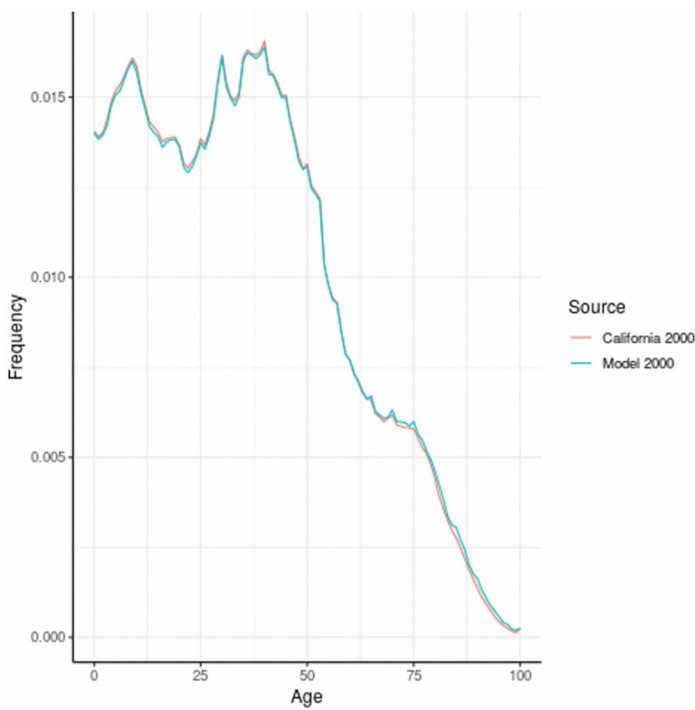

**Fig 2. Age distribution of women in California, 2000, and model based distribution.**

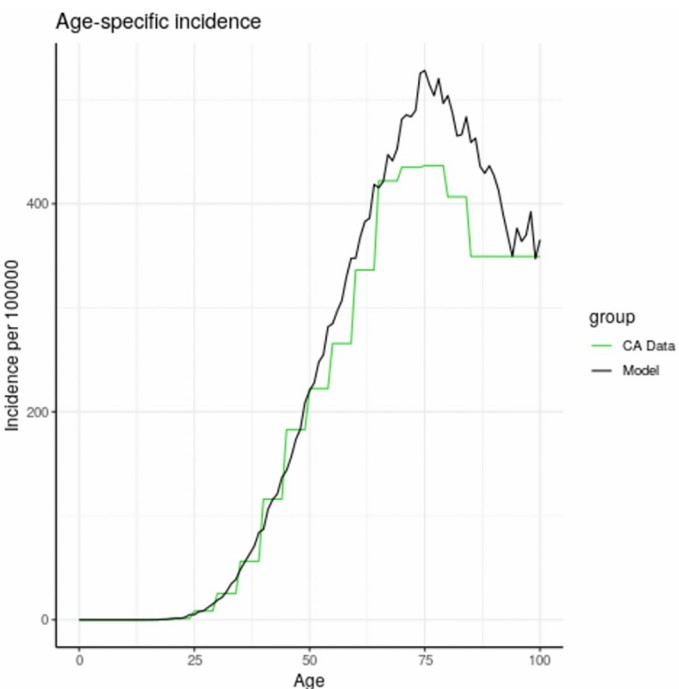

**Fig 3. Age specific incidence of breast cancer in California (2008–2012) and in model population.** The flat line at high ages (green curve) is due to the lack of specific age category information for ages 80 and above.

The age specific incidence and cumulative incidence in breast cancer in California, comparing the actual data for California from 2008–2012 with our modeled population is shown in Fig 3. The model was calibrated to California Cancer Registry data (SEER). The step functions for the actual California data (green line) are due to the 5 year age categories available for this data which we obtained from SEER. The age-specific incidence curve had 3 parts—initial onset, reduced slope at later ages and a drop off after the age of 75. We next present age-specific incidence and cumulative incidence for variables included in the model. Most exposures are presented in the form of the actual incidence as a function of presence/absence of an exposure variable, for example BRCA, in the synthetic population. The exceptions are OC, HT, and screening, which are each presented as outcomes in a simulated randomized controlled trial rather than as observational outcomes in the synthetic population because they provided a comparison group for the effect of these exposures. Chemical exposures are modeled using scenarios (i.e.,II.B.8.).

## III.B. Population model

**III.B.1. Genetic exposures.** We calibrated the effect of BRCA1 to data, from multiple scenarios as described in section II.B.1. We chose the second scenario "increased accumulation of damaged cells" for our general model (Fig 4). BRCA is a DNA repair gene, and the mutation causes decreased repair of damaged DNA, increasing accumulation of damage [56]. Cumulative risk at age 75 due to BRCA is 0.57. After calibration, the model also produces a cumulative risk of 0.57 at age 75 associated with possession of the BRCA1 gene. Fig 4a shows the cumulative incidence of breast cancer by BRCA1 status. The well-known importance of this genetic trait is reflected in the markedly higher cumulative incidence and age-specific incidence. Based on our models, as shown in Fig 4b, the differential in incidence reaches its peak around the age of 60, with much less when overall incidence declines with ages.

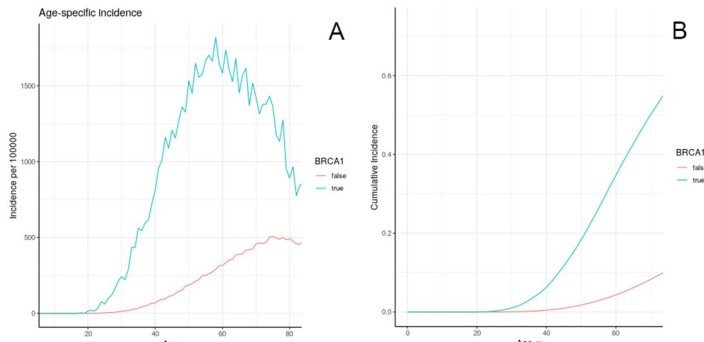

**Fig 4. Age-specific incidence (a) and cumulative incidence (b) of breast cancer in model individuals in the California/NHANES synthetic population, stratified by BRCA1 exposure.**

We next present information on genome-wide influences on age specific incidence (Fig 5a) and cumulative incidence (Fig 5b) from low-penetrance genes that have effects similar to the effect of BRAC1.

**III.B.2. Alcohol.** Alcohol exposure can increase the risk of cancer and is treated as an exogenous hazard unrelated to hormone exposure. Fig 6 shows the result of calibrating effect of alcohol consumption to breast cancer incidence. Suzuki [74] reported relative risk associated with consumption of 10g/day of alcohol at 1.12 (95% CI 1.08–1.15) for ER+ and 1.04 (95% CI 0.98–1.09) for ER-/PR- cancers, while the Collaborative Group [75] reported 7.1% increase in risk per 10g/day. We calibrated the effect of 10g/day alcohol on tissue transformation to produce comparable increase in cumulative risk of cancer to age 75 (Fig 6). The relative risk associated with moderate drinking in model individuals is 1.37. As shown in Fig 6a, we would expect cumulative incidence in California to be approximately 3% lower over all ages if women were in the low alcohol use category of 0 to 125 drinks per year.

**III.B.3. Oral contraception.** The observed relative risk associated with ever use of oral contraception was estimated at 1.24 (95% CI: 1.15–1.33). In detail, 1–4 years after stopping RR was 1.16 (95% CI: 1.08–1.23), 5–9 years after stopping was 1.07 (95% CI: 1.02–1.13), and after 10 years was effectively no longer elevated [61] (Table 1). The relative risk associated with use of oral contraception in the model was 1.17 during use, 1.17 in the first four years after use, 1.17 five to nine years after stopping, and 1.24 after ten or more years.

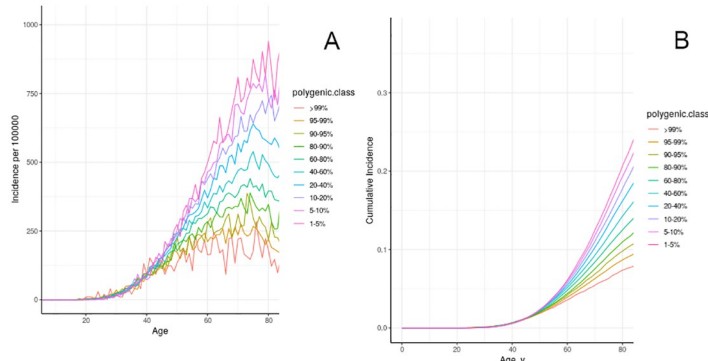

**Fig 5. Age-specific incidence (a) and cumulative incidence (b) of breast cancer in model individuals in the California/NHANES synthetic population, stratified by polygenic risk score.**

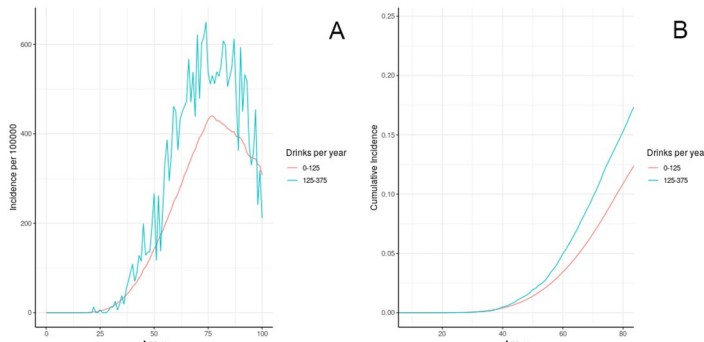

**Fig 6. Age-specific incidence (a) and cumulative incidence (b) of breast cancer in model individuals in the California/NHANES synthetic population, stratified by alcohol use.**

**Table 1. Relative risks associated with oral contraceptives compared with model results.**

| Quantity | Reported Value | Model Value |
|---|---|---|
| Relative risk during oral contraception use | 1.24 (1.15–1.33) | 1.17 |
| Relative risk 1–4 years after oral contraception use | 1.16 (1.08–1.23) | 1.17 |
| Relative risk 5–9 years after oral contraception use | 1.07 (1.02–1.13) | 1.17 |
| Relative risk 10+ years after oral contraception use | 1.01 (0.96–1.05) | 1.25 |

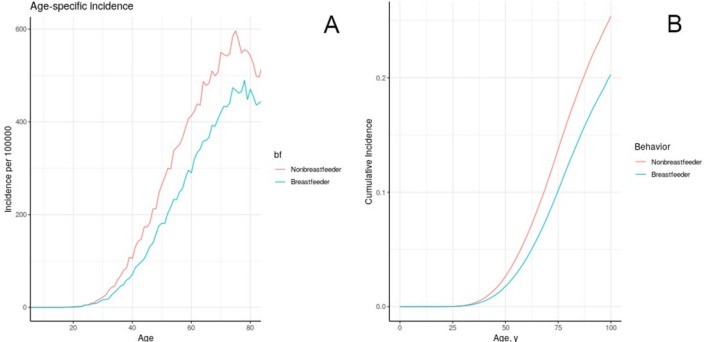

**Fig 7. Age-specific incidence (a) cumulative incidence (b) of breast cancer in model individuals in the California/NHANES synthetic population, stratified by ever breastfeeding.**

**III.B. 4. Breastfeeding.** The observed relative risk of breast cancer associated with breastfeeding has been estimated at 0.78 (95% CI: 0.64–0.94) for ER+/PR+, 0.74 (95% CI: 0.61–0.89) for ER-/PR- [48]. In the absence of clear evidence to the contrary we opted to model them as having the same effect. The effect of breastfeeding on tissue removal was calibrated to produce comparabledecrease in cumulative risk to age 75 in model individuals (Fig 7). The relative risk to age 75 associated with breastfeeding in model individuals was 0.74.

**III.B.5. Hormone therapy.** The impact of hormone therapy on tissue growth rates was calibrated to the risk results reported by the Oxford Collaborative Group [65], which were stratified by duration of use and time since last use, and controlled for a number of demographic variables. We calibrated by a simulated experiment, in which the treatment group was comprised of HT users sampled from NHANES and the control group was made up of NHANES HT users altered to not use HT (Table 2).

**Table 2. Relative risks associated with hormone therapy compared with model results.**

| Quantity | Reported Value (Confidence Interval) | Model Value |
|---|---|---|
| Relative risk of hormone therapy by duration of use: 1–4 years | 1.05 (1.011–1.089) | 1.08 |
| Relative risk of hormone therapy by duration of use: 10–14 years | 1.09 (1.003–1.177) | 1.24 |
| Relative risk of hormone therapy by time since first use: <5 years | 0.99 (0.925–1.055) | 1.04 |
| Relative risk of hormone therapy by time since first use: 5–9 years | 1.11 (1.042–1.178) | 1.14 |
| Relative risk of hormone therapy by time since first use: 10–14 years | 1.19 (1.113–1.267) | 1.11 |
| Relative risk of hormone therapy by time since first use: 15–19 years | 1.22 (1.139–1.301) | 1.16 |
| Relative risk of hormone therapy by time since first use: > = 20 years | 1.20 (1.125–1.275) | 1.10 |
| Relative risk of hormone therapy by time since last use: 1–4 years | 1.10 (1.037–1.163) | 1.08 |
| Relative risk of hormone therapy by time since last use: 5–9 years | 1.01 (0.942–1.078) | 1.17 |
| Relative risk of hormone therapy by time since last use: 10–14 years | 1.05 (0.966–1.134) | 1.17 |
| Relative risk of hormone therapy by time since last use: > = 15 years | 1.12 (1.036–1.204) | 1.08 |

**III.B.6. Chemical exposures.** The results of simulating the effects of harmful chemical exposures on breast cancer incidence in an idealized, uniform population, under a number of assumptions about the effect of exposure on physiology, are depicted in Figs 8–12. These results illustrate multiple possible actions of environmental chemicals on breast cancer incidence.Note that we can compare the timing of incidence resulting from these exposures, but in general the magnitude of the outcome should not be compared, since the exposures have not been calibrated to produce comparable magnitudes of outcome. Exceptions are: 1-year introductions of damaged tissue (including the prenatal case) can be compared for magnitude, and 10-year introductions of damaged tissue can be compared for magnitude.

**III.B. 9. Screening.** The effect of cancer screening on incidence was evaluated in the model by simulation of the study reported by Miller[51–53]. In this case the model results did not closely reproduce the reported results as summarized in Table 3.

## III.C. Sensitivity analysis

Results of sensitivity analysis are displayed in Fig 13. Each shaded square represents the partial correlation coefficient between a model parameter and an outcome with the increase or decrease in an outcome variable associated with a unit of increase in a parameter value. Many of these sensitivity values are relatively close to zero, while a few larger correlations stand out. We note that the parameter "growth_rate_attenuation" controls the gradual increase in tissue growth rates at later classess of transformation relative to the first stage, and that its class appears to have an especially large impact on outcomes having to do with the time lag from exposures to cancer incidence. The relation between BMI and some of the same outcomes may be worthy of further investigation in the future.

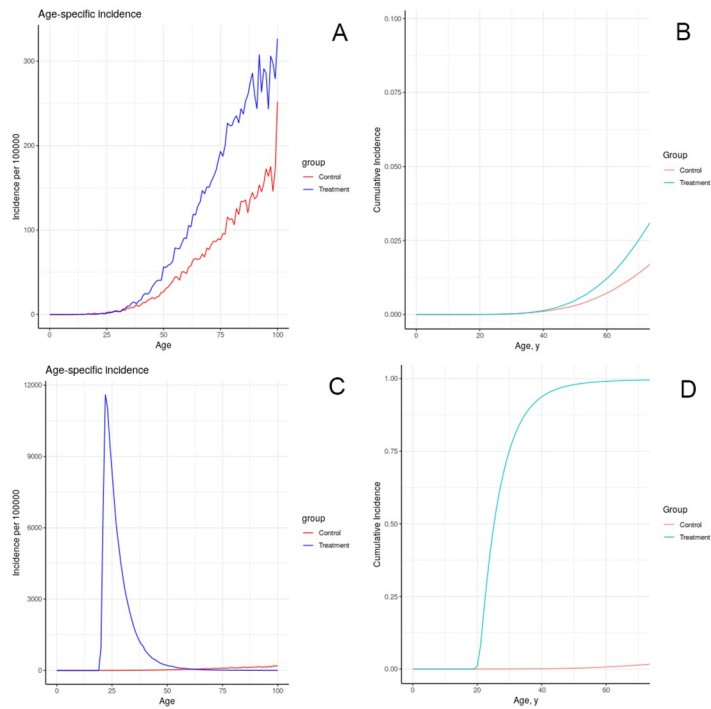

**Fig 8. Effect of brief increase in damage on simulated incidence of (A, B) advancing a portion of each individual's tissue to the first class of damage at age 20; (C, D) advancing a portion of each individual's tissue to the fifth class of damage at age 20.**

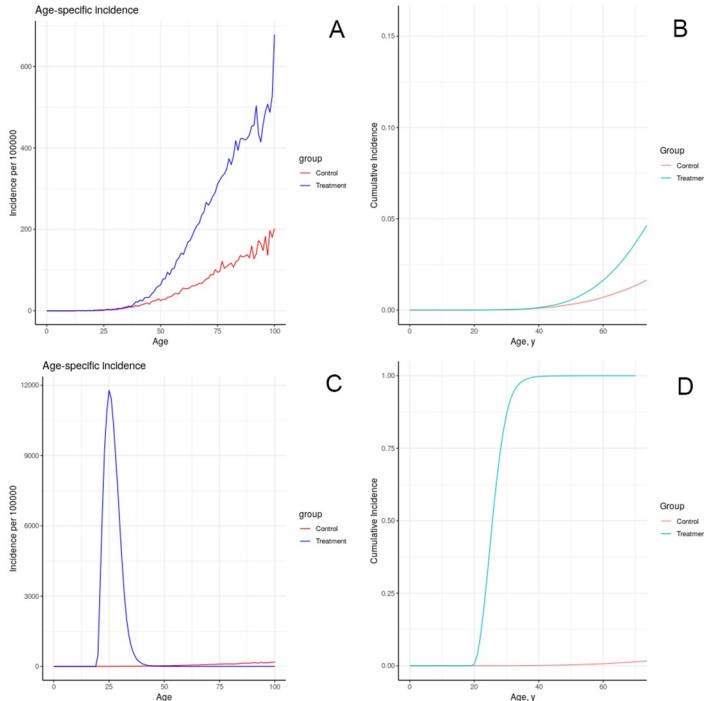

**Fig 9. Effect of sustained increase in damage on simulated incidence of (A, B) advancing a portion of each individual's tissue (half as much as in previous figure) to the first class of damage each year for 10 years starting at age 20; (C, D) advancing a portion of each individual's tissue to the fifth class of damage each year as in (A, B).**

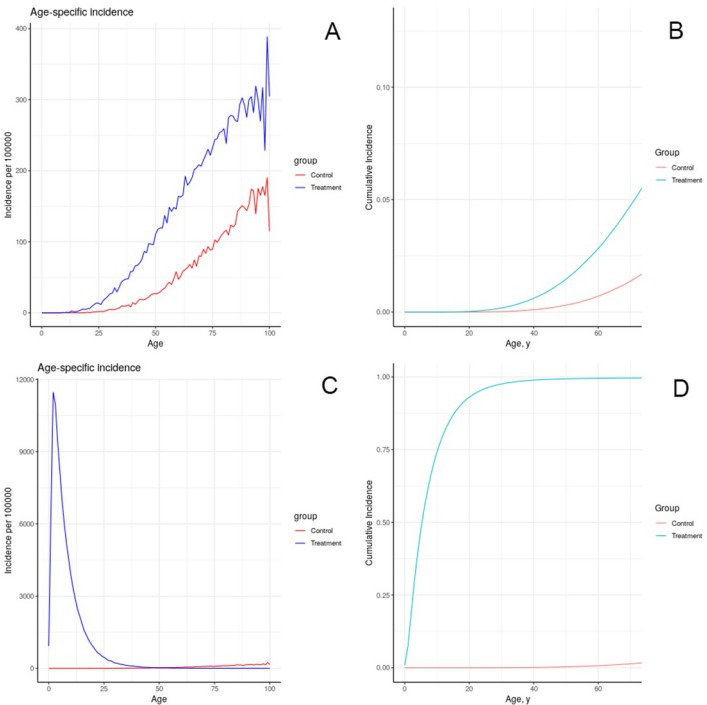

**Fig 10. Effect of prenatal increase in damage on simulated incidence of (A, B) advancing a portion of each individual's tissue to the first class of damage at birth; (C, D) advancing a portion of each individual's tissue to the fifth class of damage at birth.**

## IV. Discussions/Conclusions

Most models of breast cancer seek to predict outcomes for individual women [17,19,22], to predict the population impact of interventions at the clincal level (e.g., screening and treatment) [23,24] or to explain aspects of the biology of breast cancer, such as the process of metastasis[25]. The PII model attempts to describe breast cancer in a population, in this case the population of women in the State of California. The PII model is one example of a systems epidemiology approach, which has been defined as the study of "risk and outcomes that incorporates high-dimensional measurements from multiple domains, assesses the inter-relationships between risk factors, and considers changes over time" [76]. This perspective is of value to

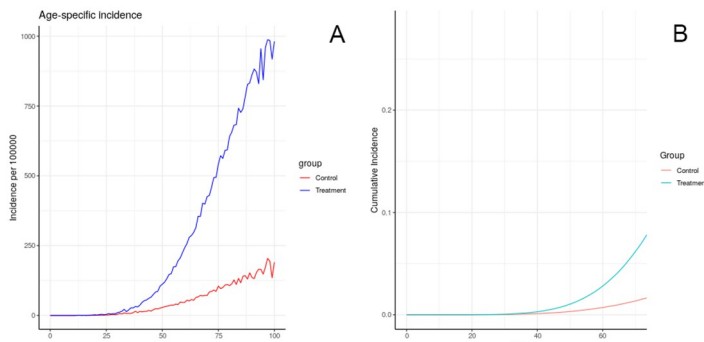

**Fig 11. Effect of lifelong increase in tissue growth rate on simulated incidence of increased tissue growth rate over the entire life course.**

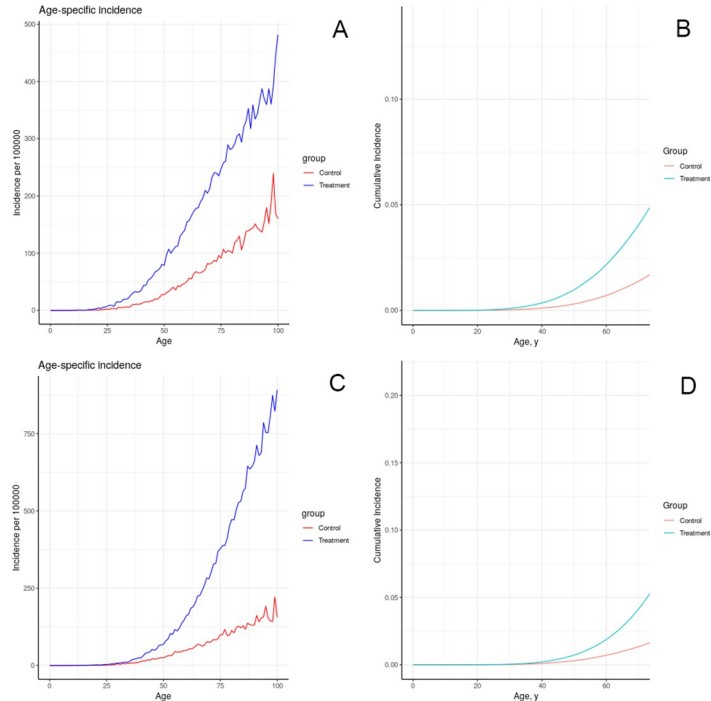

**Fig 12. Effect of lifelong decrease in tissue removal rate on simulated incidence of (A, B) advancing a portion of each individual's tissue to the first class of damage at birth; (C, D) advancing a portion of each individual's tissue to the fifth class of damage at birth.**

public health investigators and decision-makers interested in the impact of breast cancer on population health. We use a risk modeling approach, synthesizing demography and life course events, calibrated to observed incidence data and informed by studies of risk factors. The PII model illustrates age specific patterns, demonstrates the effects of changes from tissue level to individual to population, including the effects of decreasing breast density, BRCA1 genetic status, exogenous hormonal exposures, alcohol and smoking exposures, the impact of chemical toxicologic exposures of a number of types and changes in population-based screening guidelines. The PII model views women as "agents" who have individual physiologic characteristics that determine the growth, transformation and removal of breast tissue over their lifetimes and then, at the population level, how exposures from behaviors (e.g., alcohol consumption) and the environment (e.g., environmental chemicals, mammographic screening) influence the evolution of pathophysiologic processes that can result in the diagnosis of a breast cancer.

The PII model recognizes that breast cancer etiology is a complex process and tries to incorporate influences from multiple levels of determinants from "genes to society". A limitation of current approaches to modeling breast cancer, including the PII model, is that the model must

**Table 3. Relative risks in screening study compared with model results.**

| Quantity | Reported Value | Model Value |
|---|---|---|
| Relative risk in first year of screening age 40–49 | 3.1 | 1.81 |
| Relative risk in second-fifth year of screening age 40–49 | 1.25 | 1.03 |
| Relative risk in first year of screening age 50–59 | 3.6 | 1.74 |
| Relative risk in second-fifth year of screening age 50–59 | 1.25 | 1.02 |

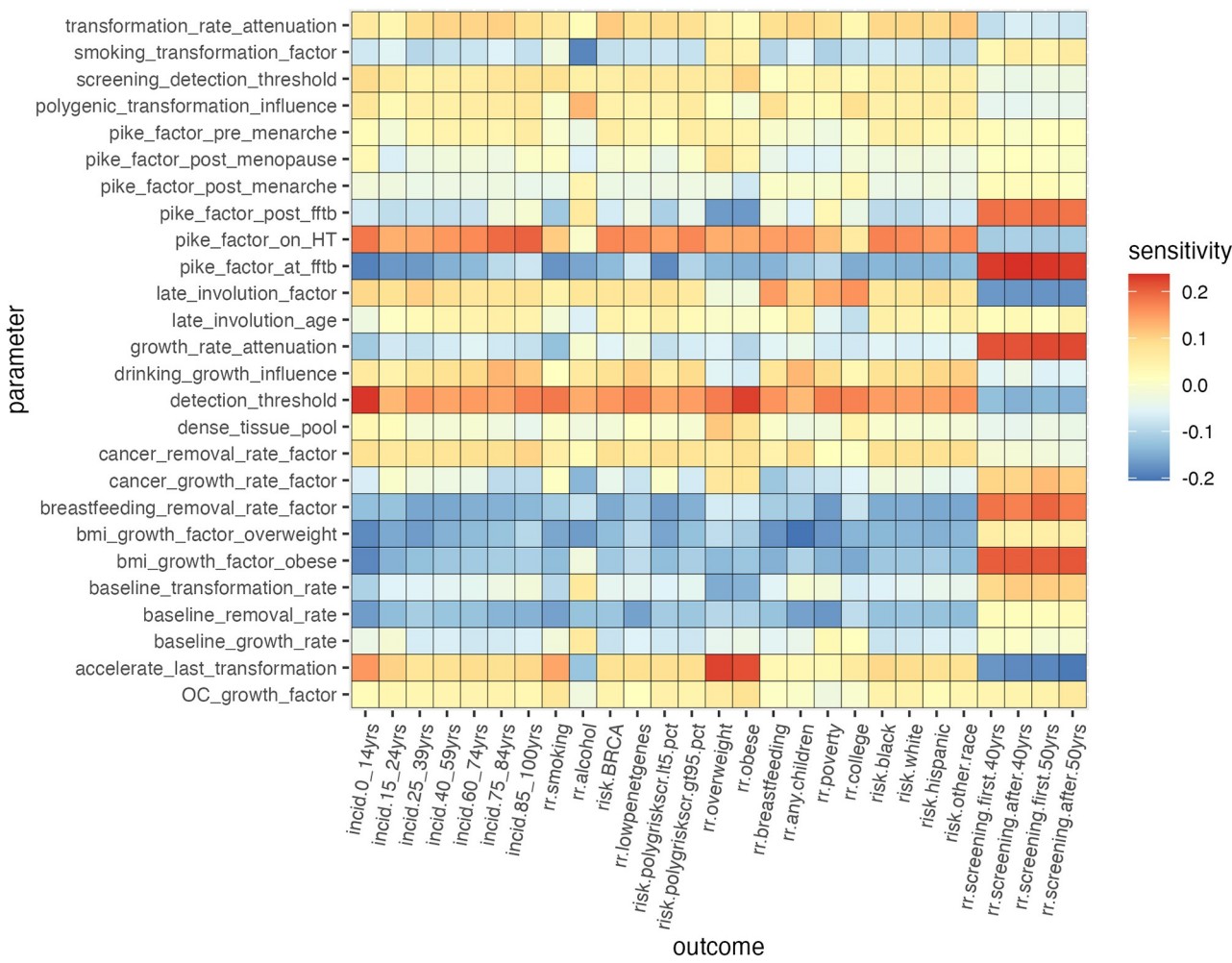

**Fig 13. Sensitivity of absolute and relative risks to model parameters as reflected by their partial correlation coefficients.**

be limited in scope to avoid being overly cumbersome, but still reflect breast cancer incidence in California. Also as the complexity of models increase, overfitting may result and generate biases that make generalization difficult [77]. For example, one simplication was to assume that individuals coded as smokers were smokers throughout their lives (lns 359–360). Future interations of the PII model can assume that some fraction of the ever smolers became ex-smokers. Future explorations can simulate effects of alterations in malleable factors affecting behaviors (e.g., dietary and alcohol consumption, effects on obesity), regulation of environmental chemical exposures (e.g., air pollution) and social stressors (e.g., income inequality).

The PII model qualitatively replicates the observed pattern of age-specific incidence, including the observed drop at higher ages. The current model overestimated breast cancer incidence in women over 75 possibly due to the small absolute numbers of cases over that age. The fit in ages over 75 years may be improved by additional calibration. The model includes the influence of multiple developmental and reproductive factors such as menarche, breast density, breastfeeding, and menopause as well as the effects of genetics for both high and low penetrance genes, race/ethnicity in the population of California women and obesity. Behaviors included alcohol consumption, smoking, exogenous estrogen use (i.e., oral contraceptives and hormone therapy) and screening with mammography. We note that the model did not closely

reproduce the values for screening generated from the cited publications of the Canadian National Breast Screening study [51–53] and we do not currently have an explanation for this discrepancy. The model also incorporates estimates of the potential impact of environmental chemicals in early development and adulthood. All these factors interact within the model to reflect their real-life causal nature rather than the more traditional approach using multivariable regression that largely focuses on the independent influence of individual factors. Our approach begins to treat a complex system with a flexible modeling framework that more accurately represents how multiple factors interact and change over the life course.

The strengths of the PII model derive from its population health perspective and value to decision-making at the policy level. The simulation will hopefully allow future investigations to fill gaps by providing a framework to understand the areas where more empirical data are needed. in our knowledge and help the lay public understand the complexity of this most common of cancers. Among its limitations are that its representation of some risk factors are approximations and the transformation rates do not necessarily correspond to specific biologic processes or observable clinical states. We did not attempt to subdivide breast cancer into the several distinguishable subtypes based on biomarkers now accepted in the field and it remains an open challenge to find ways to incorporate the finding from animal studies into a human agent based model. Our results may be made to more accurately reflect known relationships with further calibration. Its value in solving outstanding questions in breast cancer etiology has yet to be demonstrated. Our intent is to make the code for the PII model available to interested investigators [78].

The goals of simplicity and parsimony in modeling are balanced against the need to represent the age-specific incidence of breast cancer in light of the many known risk factors, as well as socioeconomic and race/ethnic disparities. In principle, the model can be applied to any population, and can be used to make prospective forecasts of incidence. Our model provides a path to integrate population health data and prospective studies in constructing a unified view of this complex disease.

The process of arriving at the PII model was truly transdisciplinary with input from a team of scientists, community representatives, and modelers with expertise in diverse fields including genetics, epidemiology, biostatistics, and social sciences. Input was invited and received in open collegial interactions over a two-year period with multiple iterations of results and modifications to the model. As with other approaches to complex systems problems, we found a transdisciplinary approach to be essential to the development of the PII model [79].

Modifications will continue as further attempts are made to refine the model, use it for addressing particular problems or challenges in population health, and to add additional factors as science presents them. Although there remain many limitations, the refinement of the model is an iterative process. Finally, the PII model did accomplish our goal of creating a model to explore the role of more upstream determinants at the population level while illustrating the contribution to breast cancer causation of multiple factors at different levels of biologic organization.

## Supporting information

**S1 Appendix.**
(DOCX)

## Author Contributions

**Conceptualization:** Robert A. Hiatt, Lee Worden, David Rehkopf, Natalie Engmann, Melissa Troester, Sarah Gehlert, Ross A. Hammond, George Kaplan, John Kornak, Krisida Nishioka, Thomas McKone, Martyn T. Smith, Leonardo Trasande, Travis C. Porco.

**Data curation:** Robert A. Hiatt, Lee Worden, David Rehkopf, Natalie Engmann, Kaya Balke, Suzanne E. Fenton, Travis C. Porco.

**Formal analysis:** Robert A. Hiatt, Lee Worden, Christian Jackson, Travis C. Porco.

**Funding acquisition:** Robert A. Hiatt.

**Investigation:** Robert A. Hiatt, David Rehkopf, Natalie Engmann, Melissa Troester, John S. Witte, Janice Barlow, Suzanne E. Fenton, Sarah Gehlert, Ross A. Hammond, George Kaplan, John Kornak, Krisida Nishioka, Thomas McKone, Martyn T. Smith, Leonardo Trasande, Travis C. Porco.

**Methodology:** Robert A. Hiatt, Lee Worden, David Rehkopf, Natalie Engmann, Melissa Troester, John S. Witte, Suzanne E. Fenton, Sarah Gehlert, Ross A. Hammond, George Kaplan, John Kornak, Krisida Nishioka, Thomas McKone, Martyn T. Smith, Leonardo Trasande, Travis C. Porco.

**Project administration:** Robert A. Hiatt, Kaya Balke.

**Resources:** Robert A. Hiatt, David Rehkopf, Suzanne E. Fenton, Travis C. Porco.

**Software:** Lee Worden, Travis C. Porco.

**Supervision:** Robert A. Hiatt, Kaya Balke, Travis C. Porco.

**Validation:** Lee Worden.

**Visualization:** Christian Jackson.

**Writing – original draft:** Robert A. Hiatt, Lee Worden, Travis C. Porco.

**Writing – review & editing:** Robert A. Hiatt, Lee Worden, David Rehkopf, Natalie Engmann, Melissa Troester, John S. Witte, Kaya Balke, Janice Barlow, Suzanne E. Fenton, Sarah Gehlert, Ross A. Hammond, George Kaplan, John Kornak, Krisida Nishioka, Thomas McKone, Martyn T. Smith, Leonardo Trasande, Travis C. Porco.

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
