## [Decision Letter · Decision Letter 0]

11 May 2022

PONE-D-21-01507A complex systems model of breast cancer etiology: The Paradigm II Model.PLOS ONE

Dear Dr. Hiatt,

Thank you for submitting your manuscript to PLOS ONE; I sincerely apologise for the unusually delayed review timeframe. After careful consideration, we feel that it has merit but does not fully meet PLOS ONE’s publication criteria as it currently stands. Therefore, we invite you to submit a revised version of the manuscript that addresses the points raised during the review process. Your manuscript has been assessed by two reviewers, whose comments are appended below. Both reviewers comment positively on the interesting and important approach. However, they both raise concerns regarding the methodological detail and apparent discrepancies between outputs suggested by the model and real-life incidences of breast cancer, among other issues. These points must be addressed by appropriate revisions.

Again, I sincerely apologise for the delay in sending this decision. We look forward to receiving your revised manuscript.

Kind regards,

Emily Chenette

Editor in Chief

PLOS ONE

**Journal requirements:**

“No authors have competing interests.”

“R.A.H. California Breast Cancer Research Program (20ZB-8303).”

“This work was supported by the California Breast Cancer Research Program (20ZB-8303 to R.A. Hiatt).”

“R.A.H. California Breast Cancer Research Program (20ZB-8303).”

6. Please include your tables as part of your main manuscript and remove the individual files. Please note that supplementary tables (should remain/ be uploaded) as separate ""supporting information"" files.

Reviewers' comments:

Reviewer's Responses to Questions

**Comments to the Author**

1. Is the manuscript technically sound, and do the data support the conclusions?

Reviewer #1: Yes

Reviewer #2: Partly

2. Has the statistical analysis been performed appropriately and rigorously? 

Reviewer #1: N/A

Reviewer #2: N/A

3. Have the authors made all data underlying the findings in their manuscript fully available?

Reviewer #1: Yes

Reviewer #2: No

4. Is the manuscript presented in an intelligible fashion and written in standard English?

Reviewer #1: Yes

Reviewer #2: Yes

5. Review Comments to the Author

Reviewer #1: In the manuscript entitled “A complex systems model of breast cancer etiology: The Paradigm II Model”, Hiatt et al developed an agent-based model using a systematic epidemiology approach to explore determinants of breast cancer at both population and biologic levels. The model was expected to illustrate the complex etiology of breast cancer and to make prospective forecast of incidence. The current model, as mentioned by the authors, may reflect real-life causal nature rather than the traditional multivariable regression approach that focuses on the independent influence of individual factors. The topic is intriguing, but several issues/limitations should be noted.

1. The authors constructed idealized population by simulating individuals based on multiple sources (data sources from different countries, results from different studies, etc.) and assumptions (e.g., Line 207, “we began with a baseline growth rate assumed to be the same for all stages of modified tissue except the final stage). How did they prove/confirm that those simulations or assumptions were valid? The authors may have already cited references, but some clarifications in the paper may be helpful in understanding the methodology behind the model.

2. This model seems more complex than the traditional models and involves data across multiple levels. Lack of generalizability may be an issue as the authors may have noted already. Another question is how feasible is it to use this model in the real-world setting/in practice to predict incidence or help in public health decision-making as the authors suggested?

3. According to the sensitivity analysis, some variables added little value to the model. Please explain the rationale for variable selection (why the authors decided to keep those variables in the model even though they did not contribute much).

Some minor points:

1. There is no “III A” heading prior to “III B”.

2. In Table 3, the model values were quite different from the reported values. Please explain.

3. Line 301, please add reference for this pooled study.

Reviewer #2: FORMATTED VERSION PROVIEDED AS ATTACHMENT

The authors should be commended for their ambitious and endeavor to create a complex systems population-oriented model of breast cancer incidence. Making a model like this represents a substantial undertaking, and the authors have wisely assembled an excellent transdisciplinary team. I believe the Paradigm II model has the potential to be a useful population health tool. However, I think that the current manuscript needs substantial work to achieve the goals of 1) understandability by a broad readership, 2) transparency, and 3) making a strong case that the models outputs are trustworthy. I provide a point-by-point critique below, but general themes include some ambiguous language in the Methods section, a lack of rationale for certain design choices, a lack of framing for how/why the results presented demonstrate the utility/credibility of the model, and a lack of explanations for discrepancies between modeled and observed results based on the same data sources. I believe that providing more methodological detail in one or more additional appendices would allow the authors to make the Methods section more readable while including the necessary detail needed for transparency/reproducibility.

Abstract

- Some empirical metrics in the Abstract seem in order.

- Line 59 – ‘The resulting model closely replicates most risk factor distributions’ does not seem capture the type of results presented. The results section of the paper presents mostly summaries of how the model reproduced incidence or relative risk due to specific risk factors.

Introduction

- Lines 123-126 – The placement of this language, and the phrasing, may suggest to some that the manuscript is going to address the enumerated questions. I think it is important to illustrate the rationale for developing the PII model, but I suggest rephrasing to signal to the reader that you are talking about future applications of the model.

Methods

- Generally, some detail is needed about the specific calibration approaches used.

- 140 – A bit of detail is needed about the ‘previously described methods’.

- 147 – The concept of ‘removal’ needs to be explained clearly early on.

- 161-162 – A reference is needed for this statement.

- 173- Drop first ‘incliuded’

- 180 – I found the sentence beginning at line 180 quite unclear.

- 191 – Does ‘healthy tissue’ refer to ‘healthy dense tissue’? Is non-dense tissue relevant to the events of interest in the model?

- 225-231 – Is total breast tissue relevant, or just dense breast tissue?

- 226 – What is meant by ‘others’? Does this refer to subsequent modified tissue sates?

- 238-240 – It Is said that both breast density and the ‘effects of breast density’ are a function of dense volume and BMI. Is there a distinction in meaning between these statements. If so, that distinction needs to be better clarified. If not, the redundancy needs to be eliminated.

- 240-242 – Is this statement not redundant of the statement in 235-237?

- 255-256 – it is a minor limitation that polygenic risk score was not conditioned on any other variables like race.

- 277 – The meaning of the phrase ‘risk data’ is not clear. Does this refer to cumulative incidence?

- 271-277 – The authors may want to briefly mention the rationale for directly considering only BRCA1 and not BRCS 2.

- 311-315 – I find this sentence confusing. Is one of the points that start/stop years were specified relative to menarche and menopause?

- ‘Stage’ is used to describe different model states stemming from cell damage in the early part of the paper (e.g. lines 154 and 191). Is ‘class’ as used on line 534 intended to convey the same concept? How does this concept different from ‘classes of damage’ used throughout the latter part of the paper?

- 353-358 – I found these sentences quite confusing. The rationale for choosing the particular rules for damage needs to be better explained. I think that a figure tied to the Methods section (perhaps a more detailed Figure 1) could help convey the concept of transitions and the types of damage relevant to each.

- 358-362 – Is this describing a calibration approach? Please clarify.

Results

- Figure 3 – The model seems to substantially overestimate incidence in women in their 70’s.

- Figure 13 – This figure is very difficult to read. Some more descriptive variable labels are needed for the horizontal (outcomes) axis. What are the units represented by the scale?

- 394-397 – Please explain the rationale for using simulated RCTs as the basis for analysis of OC, HT, and screening as an alternative to stratifying observed data from the synthetic population.

- III.B.3 / Table 1 – The same end points used to calibrate the PII model (Reference 53) appear to be used as comparators to the modeled outputs in Table 1. I am not sure of the value that this exercise provides, particularly given that the modeled and observed effect estimates do not correspond well.

- III.B.5 / Table 2 – A similar critique can be applied as for the authors’ treatment of oral contraceptive exposure (III.B.3 / Table 1) discussed above.

- III.B.9 / Table 3 – A similar critique can be applied as for the authors’ treatment of oral contraceptive exposure (III.B.3 / Table 1) discussed above.

Discussion

- 486-488 – I think it would be helpful to the reader in understanding the ‘big picture’ to bring out this point more strongly in the introduction.

- Generally, I think it would be helpful to provide some pros and cons of PII relative to the CISNET models of breast cancer incidence and screening.

6. PLOS authors have the option to publish the peer review history of their article (what does this mean?). If published, this will include your full peer review and any attached files.

Reviewer #1: No

Reviewer #2: **Yes: **Johnie Rose, MD, PhD

---

## [Author Response · Author response to Decision Letter 0]

5 Feb 2023

We have responded to each point in their critiques below. When referring to where changes have been made in the revised manuscript, we note the new lines in the Track Changes version not the clean version.

Review #1

1.How did they prove/confirm that those simulations or assumptions were valid? The authors may have already cited references, but some clarifications in the paper may be helpful in understanding the methodology behind the model. 

Thank you. The paper is meant to describe the development of the Paradigm II model and the whole paper describes the methodology, but in this revision, we clarify our goal (new lns 109-111) and our intention to test the model for a number of different scenarios as mentioned in new lns. 134-137.

2. Lack of generalizability may be an issue as the authors may have noted already.

It is true that breast cancer does not occur in a homogeneous fashion all over the world or even in the United States. However, California is a large and diverse population representing 12% of the U.S., and the current PII model attempts to reflect the determinants of new breast cancer cases in this population as previously noted in new lns 153-155

3. Another question is how feasible is it to use this model in the real-world setting/in practice to predict incidence or help in public health decision-making as the authors suggested?

Our intention is to demonstrate the feasibility of using this model in a number of real-world setting once this methods paper is published and in the public domain. Examples of its application for important public health issues are given in new lns 134-137.

4. According to the sensitivity analysis, some variables added little value to the model. Please explain the rationale for variable selection (why the authors decided to keep those variables in the model even though they did not contribute much).

We wanted to keep all the variables in the model to illustrate the contribution of each of them, including some which added little value. We felt that that was useful information to include. Some variables/factors may be important at an individual level to predict breast cancer, but not at a population level.

Minor Points

5. There is no “III A” heading prior to “III B”. 

Thank you. ‘III.A. Physiology Model’ has been added (new ln 418)

6. In Table 3, the model values were quite different from the reported values. Please explain.

We agree that the model values are different that the reported values but cannot currently explain why. This anomaly has been noted in new lns 489-490, 533-536

7. Line 301, please add reference for this pooled study.

Thank you for catching that oversight. We have added the citation to the meta-analysis that generated these risk levels by level of alcohol consumption. (new lns 323-326)

Review #2

Thank you for the laudatory comments. We will try to respond to the general concerns as stated: 1) understandability by a broad readership, 2) transparency, and 3) making a strong case that the model’s outputs are trustworthy.

Abstract:

1. Some empirical metrics in the Abstract seem in order.

We now provide reference to the risk factors and metrics produced from the model in the Abstract, new lns 59-62.

2. - Line 59 – ‘The resulting model closely replicates most risk factor distributions’ does not seem capture the type of results presented. The results section of the paper presents mostly summaries of how the model reproduced incidence or relative risk due to specific risk factors

We modified the language to reflect what the model reproduced for this publication, new lns 59-62.

Introduction:

3. I think it is important to illustrate the rationale for developing the PII model, but I suggest rephrasing to signal to the reader that you are talking about future applications of the model.

Thank you. We have tried to be more explicit about the rationale (new lns 109-111) and intent for future use (new lns 134-137) of the PII model.

Methods:

4. 140 – A bit of detail is needed about the ‘previously described methods’-

We hope our Methods sections describes the process we used to develop the Paradigm II model. We have elaborated on what we mean by ‘previously described methods’ in new lns 139-140. This refers to the approach detailed in our previous two publications on the conceptual aspects of this model, but our summary in the current manuscript covers the key points.

5. 147 – The concept of ‘removal’ needs to be explained clearly early on.

We see that the word ‘removal’ may not have been clear. We are referring to the process of tissue or cell loss during breast feeding, aging or surgical removal and have substituted ‘cell loss’ throughout and in Figure 1. 

6. - 161-162 – A reference is needed for this statement.

We have treated dense breast tissue as the critical target for carcinogenesis and provided references to support that contention in new ln 174. (Ursin et al. 2005; Stone et al. 2010; Engmann et al. 2019)

7. - 173- Drop first ‘included’

Done.

8. - 180 – I found the sentence beginning at line 180 quite unclear.

Reference (new ln 193) to Pike model and ‘tissue age’ – has been revised to read “This involved the idea of a transformation of “tissue age”, which advances at a time-varying rate.” which refers directly to ref 45 in the revision.

9. - 191 – Does ‘healthy tissue’ refer to ‘healthy dense tissue’? Is non-dense tissue relevant to the events of interest in the model?

Although there is interaction between ductal tissue and surrounding stroma in breast development and breast cancer, [Wiseman and Werb, 2002 – new ref 47] we have treated the healthy tissues contributing to radiologic breast density as the critical target. We have clarified this point, new lns 204-207.

10. - 225-231 – Is total breast tissue relevant, or just dense breast tissue?

Just dense tissue, now specified in new lns 204-207.

11. - 226 – What is meant by ‘others’? Does this refer to subsequent modified tissue sates?

‘Others’ referred to accumulation of subsequent transformations. This has been reworded in new ln 242. Thank you.

12. - 238-240 – It Is said that both breast density and the ‘effects of breast density’ are a function of dense volume and BMI. Is there a distinction in meaning between these statements. If so, that distinction needs to be better clarified. If not, the redundancy needs to be eliminated.

We did not distinguish between breast density and its effects, but to clarify we have added language to the paragraph in new lns 254-257.

13. - 240-242 – Is this statement not redundant of the statement in 235-237?

Yes. Thank you for noticing. Redundancy was removed. New ln 257.

14. - 255-256 – it is a minor limitation that polygenic risk score was not conditioned on any other variables like race.

Yes, we did not apply an ancestry-specific polygenic risk score due to the limited information on genetic variants in non-European ancestry populations. This is a general limitation of polygenic risk scores, and there is substantial ongoing effort to make sure they are transferable across different ancestral populations. We note this minor limitation in our revised manuscript:

“Due to the disproportionate representation of white subjects in the studies used, we assumed the distribution of risk in non-white subjects to be comparable though it may be driven by a different distribution of underlying genetic variants.” New lns 302-305.

15. - 277 – The meaning of the phrase ‘risk data’ is not clear. Does this refer to cumulative incidence?

Thank you. We have edited that sentence to read ‘…breast cancer risk of…’ new lns. 298-299

16. - 271-277 – The authors may want to briefly mention the rationale for directly considering only BRCA1 and not BRCA 2.

We considered that the risk due to BRCA1 was illustrative of the contribution of germline mutations and choose not to include BRCA2 for relative simplicity of the model. BRCA1 confers a slightly higher risk of breast cancer and is associated with more aggressive disease in comparison with BRCA2. We added a statement to clarify this decision in new lns 290-292.

17. - 311-315 – I find this sentence confusing. Is one of the points that start/stop years were specified relative to menarche and menopause?

Thank you. We have attempted to clarify this point in new lns 335-340.

18. - ‘Stage’ is used to describe different model states stemming from cell damage in the early part of the paper (e.g. lines 154 and 191). Is ‘class’ as used on line 534 intended to convey the same concept? How does this concept different from ‘classes of damage’ used throughout the latter part of the paper?

Apologies. This inconsistency was missed in the last version before submission. We have eliminated the use of ‘stage’ as potentially being confused with the clinical stage of presentation of a tumor. Instead, we now use ‘class’ throughout to reflect the evolving development of a cancer through sequential transformation of cells. 

19. - 353-358 – I found these sentences quite confusing. The rationale for choosing the particular rules for damage needs to be better explained. I think that a figure tied to the Methods section (perhaps a more detailed Figure 1) could help convey the concept of transitions and the types of damage relevant to each.

Thank you but we decided not to try to change the basic model (Figure 1) to reflect the multiple possible pathways to maintain its simplicity. In the section on Chemical exposures (II.B.8.) we describe different types or classes of exposures, not all possible exposure scenarios, in order to demonstrate how the model would apply to these selected types. We have tried to clarify this point in new lns 378-389.

20. - 358-362 – Is this describing a calibration approach? Please clarify.

No. As described in new lns 391-395, the model is intended to simulate breast cancer incidence in an idealized population. 

Results:

21. - Figure 3 – The model seems to substantially overestimate incidence in women in their 70’s.

We agree that the model overestimates incidence of women aged 75 -100 years. The model does contain a term for involution of breast tissue in older age. We may be able to improve the fit in the future with additional calibration but the small numbers of breast cancers in this elderly age group makes the simulation more difficult. We add a comment to this effect in revised new lns 425-426.

22. - Figure 13 – This figure is very difficult to read. Some more descriptive variable labels are needed for the horizontal (outcomes) axis. What are the units represented by the scale?

Thank you. We have rerun the program to create a much clearer Figure 13 and recreated the labels along the axes. These are now part of the submitted Figures and Tables. The figure is a matrix of partial correlation coefficients to be interpreted the same as any correlation. Clarification has been added to new lns. 491-494.

23. - 394-397 – Please explain the rationale for using simulated RCTs as the basis for analysis of OC, HT, and screening as an alternative to stratifying observed data from the synthetic population.

Relative risks from exposure not available from observational data, but by simulating data from an RCT we have a comparison group. New lns 431-432.

24. - III.B.3 / Table 1 – The same end points used to calibrate the PII model (Reference 53) appear to be used as comparators to the modeled outputs in Table 1. I am not sure of the value that this exercise provides, particularly given that the modeled and observed effect estimates do not correspond well.

Thank you. The reviewer makes a point. In Tables 1-3 we found that although it fits the training data, it may not be right and will need to be further explored. We demonstrate, however, that the Paradigm II model can be calibrated and can be improved upon in the future.

25. - III.B.5 / Table 2 – A similar critique can be applied as for the authors’ treatment of oral contraceptive exposure (III.B.3 / Table 1) discussed above.

See response to 24.

26. - III.B.9 / Table 3 – A similar critique can be applied as for the authors’ treatment of oral contraceptive exposure (III.B.3 / Table 1) discussed above.

See response to 24.

Discussion:

27. - 486-488 – I think it would be helpful to the reader in understanding the ‘big picture’ to bring out this point more strongly in the introduction.

Agree. We have added language in the introduction to orient the reader to the intended ‘big picture’. New lns 109-113.

28. - Generally, I think it would be helpful to provide some pros and cons of PII relative to the CISNET models of breast cancer incidence and screening.

We agree that the CISNET models deserve to be referenced because of their impact of the effects of screening and treatment on guidelines. So we have added a section on the CISNET models with two recent references in lns 100-104 and cited these same references in the Discussion, ln 501.

---

## [Decision Letter · Decision Letter 1]

27 Feb 2023

A complex systems model of breast cancer etiology: The Paradigm II Model.

PONE-D-21-01507R1

Dear Dr. Hiatt,

We’re pleased to inform you that your manuscript has been judged scientifically suitable for publication and will be formally accepted for publication once it meets all outstanding technical requirements.

Kind regards,

Marianna De Camargo Cancela, DDS, MSc, PhD

Academic Editor

PLOS ONE

Additional Editor Comments (optional):

Please carefully review the comments made by Reviewer 2 and consider incorporating their suggestions into your revised manuscript.

This work is an important contribution to the field of cancer epidemiology, and the model provides a better understanding of how risk factors are related to each other. Congratulations on this highly relevant article!

Reviewers' comments:

Reviewer's Responses to Questions

**Comments to the Author**

1. If the authors have adequately addressed your comments raised in a previous round of review and you feel that this manuscript is now acceptable for publication, you may indicate that here to bypass the “Comments to the Author” section, enter your conflict of interest statement in the “Confidential to Editor” section, and submit your "Accept" recommendation.

Reviewer #1: All comments have been addressed

Reviewer #2: (No Response)

2. Is the manuscript technically sound, and do the data support the conclusions?

Reviewer #1: Yes

Reviewer #2: Yes

3. Has the statistical analysis been performed appropriately and rigorously? 

Reviewer #1: Yes

Reviewer #2: Yes

4. Have the authors made all data underlying the findings in their manuscript fully available?

Reviewer #1: Yes

Reviewer #2: Yes

5. Is the manuscript presented in an intelligible fashion and written in standard English?

Reviewer #1: Yes

Reviewer #2: Yes

6. Review Comments to the Author

Reviewer #1: My comments have been addressed by the authors. No further questions. Appreciate the authors' efforts in revising the manuscript.

Reviewer #2: Authors: Thank you for addressing these critiques so thoroughly. I feel the revised version provides the clarity that this important work deserves.

Johnie Rose, MD, PhD

--

Please address the following:

107: With regard to the statement, “‘but do not inform interventions and policy in population health”, I don’t think this statement can be applied to CISNET given its role in informing screening recommendations.

153 – “93,826,718 women in the state of California in 2008-2012” – Is this female person-years? The population of CA is only ~40 million. Please clarify.

242-244: ‘the size of the healthy pool of breast tissue was assigned as a constant multiple of the amount of dense breast tissue with the multiplier calibrated to yield the needed incidence pattern.’ Is this to say ‘‘at time 0”? If so, this should be clarified.

396 – Please elaborate on the definition of “slightly”.

495: Change ‘an’ to ‘a’

Please consider the following changes:

Perhaps be bolder with touting what’s special about Paradigm II! In the Introduction and Discussion, I think the unique value of a complexity approach could be drawn out a bit more. As pointed out in the paper, it can be used to test population health interventions in silico. I think it can be more strongly emphasized that this approach lends itself to understanding relative and synergistic contributions of inputs at multiple levels. Finally, I think it could be mentioned in the Discussion that the approach can provide a framework to understand the areas where more empirical data are needed.

359-360: “Individuals with a value of 1 for this variable were assumed to be smokers throughout their life-course”. In the Discussion, this could be cited as a specific limitation related to simplification. Adjusting this assumption to assume that some fraction of ever smokers were not lifetime smokers could be a useful exercise (for the future).

378 – Is there a better term than ‘treated’? ‘Experimental’?

515 – I would suggest mentioning the hypothesized reason for overestimating incidence in women over 75 here (small sample size), in the Discussion section, rather than in Results.

7. PLOS authors have the option to publish the peer review history of their article (what does this mean?). If published, this will include your full peer review and any attached files.

Reviewer #1: No

Reviewer #2: **Yes: **Johnie Rose, MD, PhD

---

## [Editor Report · Acceptance letter]

11 May 2023

PONE-D-21-01507R1 

A Complex Systems Model of Breast Cancer Etiology: The Paradigm II Model 

Dear Dr. Hiatt:

I'm pleased to inform you that your manuscript has been deemed suitable for publication in PLOS ONE. Congratulations! Your manuscript is now with our production department. 

Kind regards, 

on behalf of

Dr Marianna De Camargo Cancela 

Academic Editor

PLOS ONE